# Phytochemical Screening and Antioxidant and Cytotoxic Effects of *Acacia macrostachya*

**DOI:** 10.3390/plants10071353

**Published:** 2021-07-02

**Authors:** Hamidou Têeda Ganamé, Yssouf Karanga, Issa Tapsoba, Mario Dicato, Marc F. Diederich, Claudia Cerella, Richard Wamtinga Sawadogo

**Affiliations:** 1Laboratoire de Chimie Analytique, Environnementale et Bio-Organique (LCAEBiO), Université Joseph KI-ZERBO, Ouagadougou 03 BP 7021, Burkina Faso; teedamillions89@gmail.com (H.T.G.); yssoufkaranga@gmail.com (Y.K.); issa.tapsoba@gmail.com (I.T.); 2Laboratoire de Biologie Moléculaire et Cellulaire du Cancer (LBMCC), Hôpital Kirchberg, L-2540 Luxembourg, Luxembourg; dicato.mario@chl.lu; 3Laboratoire de Chimie des Matériaux et de l’Environnement (LCME), Université Norbert ZONGO, Avce Maurice Yameogo, Koudougou BP 376, Burkina Faso; 4College of Pharmacy, Seoul National University, Seoul 08826, Korea; marcdiederich@snu.ac.kr; 5Institut de Recherche en Sciences de la Santé (IRSS/CNRST), Ouagadougou 03 BP 7192, Burkina Faso

**Keywords:** *Acacia macrostachya*, antioxidant, cytotoxicity, cancer, leukemia, lymphoma

## Abstract

*Acacia macrostachya* is used in Burkina Faso folk medicine for the treatment of inflammation and cancer. The purpose of this study was to evaluate the antioxidant and cytotoxic effects of this plant. The cytotoxic effects of root (dichloromethane **B1** and methanol **B2**) and stem (dichloromethane **B3** and methanol **B4**) bark extracts of *A. macrostachya* were assessed on chronic K562 and acute U937 myeloid leukemia cancer cells using trypan blue, Hoechst, and MitoTracker Red staining methods. The antioxidant content of extracts was evaluated using DPPH (2,2-diphenyl-1-picryl-hydrazyl) and FRAP (ferric reducing antioxidant power) methods. The root bark extracts **B1** and **B2** of *A. macrostachya* demonstrated higher cytotoxicity with IC_50_ values in a low µg/mL range on both U937 and K562 cells, while the stem bark **B4** extract selectively affected U937 cells. Overall, healthy proliferating peripheral blood mononuclear cells (pPBMCs) were not or barely impacted in the range of concentrations cytotoxic to cancer cells. In addition, *A. macrostachya* exhibited significant antioxidant content with 646.06 and 428.08 µg ET/mg of extract for the **B4** and **B2** extracts, respectively. Phytochemical screening showed the presence of flavonoids, tannins, alkaloids, and terpenoids/steroids. The results of this study highlight the interest of *A. macrostachya* extracts for the isolation of anticancer molecules.

## 1. Introduction

Cancer is a major public health problem worldwide. This disease is characterized by the uncontrolled proliferation of one or more clones of transformed cells and their destructive dissemination to the whole organism by local invasion and systemic spread [1]. To date, the methods of patient care are chemotherapy, radiotherapy, and surgery [2]. However, these methods are not always available in developing countries due to their very high cost, lack of adequate equipment, and qualified medical staff. In these countries, more than 80% of people use traditional medicine for primary health care [3]. In this context, products derived from medicinal plants may be an alternative in searching for new anticancer molecules. Indeed, between 1983 and 1994, more than 60% of cancer drugs approved in the United States of America were of natural origin [4]. Numerous studies have shown that oxidative stress is involved in the pathogenesis of many chronic non-communicable diseases, including cancer [5,6]. Free radicals are potential carcinogens because they facilitate mutagenesis, tumor promotion, and spread [7]. In addition, experiments on the role of reactive oxygen species in tumor initiation and progression have assumed that oxidative stress acts as an agent damaging DNA [8]; this would effectively increase the rate of mutation in cells and therefore lead to errors in protein manufacturing, genome instability, and cell proliferation [9]. More recent studies have shown that in addition to inducing genomic instability, reactive oxygen species can specifically activate specific signaling pathways and thus contribute to tumor development by regulating cell proliferation, angiogenesis, and metastasis. Thus, the medical community realizes that an increase in oxidative stress is potentially a cause of the development of various pathologies, such as cancer. As a result, several clinical trials have been conducted on antioxidants as cancer prevention agents [10]. The use of antioxidants by cancer patients is particularly important to maintain the balance between the free radicals generated and the trapping abilities of radicals and treat long-term complications that may occur. Scientific research has shown that plants are small pharmaceutical factories capable of producing substances with interesting biological properties, such as antioxidant properties. However, for a better appreciation of the antioxidant properties of a plant extract, the investigation of interactions between natural antioxidants and other food matrix components represents a main step [11].

In Burkina Faso, *Acacia macrostachya* (Mimosaceae) is a medicinal plant that therapists use traditionally and has interesting biological potential, such as anticancer and anti-inflammatory activities [2,12]. Indeed, the different organs of the plant are used alone or in combination with other plants in traditional medicine to treat various pathologies related to oxidative stress, such as malaria, inflammation, and muscle pain [13]. In addition, *A. macrostachya* seed extract has biological properties that contribute to the strength and tightness of the barrier function while helping to maintain a good level of skin hydration by limiting water losses [14]. It has also been shown that *A. macrostachya* seed extract stimulates the expression of heme oxygenase-1, which is part of the defense system. This thus helps to strengthen the cell’s defense and protection system from oxidative stress [14]. Furthermore, the extract helps protect cells from age-related alterations [14]. Of note, investigations of leaf, stem, and root bark extracts of *A. macrostachya* revealed the presence of secondary metabolites, such as saponins, flavonoids, tannins, and alkaloid salts [13,15]. Furthermore, qualitative analyses of methanol extracts from the roots of this plant revealed the presence of triterpenes, steroids, and tannins [2,16]. The presence of these secondary metabolites could justify the antioxidant and anticancer properties [2] of this plant. Methanol extracts from the roots of *A. macrostachya* indeed induced significant inhibition of the proliferation of the cancerous cell line KB, with less effect against normal Vero and MRC-5 cells [2]. Furthermore, these same extracts are nongenotoxic and strongly trap DPPH^●^ radicals with an IC_50_ of 4.30 ± 0.26 µg/mL [2]. Methanol–water extracts (1:1 *v*/*v*) have significant antiplasmodial activity, while methanol and dichloromethane extracts significantly trap DPPH^●^ radicals with IC_50_ values of 11.40 ± 2.08 µg/mL and 14.92 ± 1.39 μg/mL, respectively [13]. Coulibaly et al. recently showed that dichloromethane extract from *A. macrostachya* root bark contains molecules with relevant anti-inflammatory and analgesic properties [15,17].

Given the above and to contribute to the isolation of new anticancer molecules, the study of the cytotoxicity of *A. macrostachya* extracts appears necessary.

In this work, the antioxidant and cytotoxic effects of root and stem bark extracts of *A. macrostachya* were investigated in chronic (K562) and acute (U937) myeloid leukemia cell models. In addition, the phytochemical compounds of these extracts were also explored.

## 2. Results

### 2.1. Phytochemical Screening

Qualitative analysis by thin-layer chromatography (TLC) allowed us to highlight the ubiquitous presence of terpenoids/steroids. In addition, flavonoids and tannins were present or not depending on the extract; alkaloids, instead, were found only in the macerated root bark (**B1**) extract (Appendix A). Colorimetric tests were also used to support TLC results. The same procedure was followed for all the extracts. It emerges from the results of the phytochemical study that methanol extracts from the root and stem barks of *A. macrostachya* (**B2** and **B4**, respectively) contain flavonoids, tannins, and terpenoids/steroids, while alkaloids are absent. For dichloromethane extracts, we noted the presence of flavonoids, terpenoids/steroids, and alkaloids in the macerated root bark (**B1**), while in **B3**, only terpenoids/steroids were highlighted. On the other hand, tannins are absent in these two extracts. We also noted that only **B1** contains alkaloids.

### 2.2. Total Phenolic Compound, Flavonoid, and Tannin Contents

The values of TPC expressed in µg equivalent of gallic acid (µg EGA), TFC expressed in µg equivalent of quercetin (µg EQ), and TCT expressed in µg equivalent of catechin (µg EC) per mg of raw extract are grouped in Figure 1.

Figure 1 shows that families and subfamilies of phenolic compounds from methanol extracts of *A. macrostachya* are at remarkable proportions. Indeed, **B2** and **B4** contained the highest polyphenol levels, estimated at 210.183 and 280.965 µg EGA/mg extract, respectively. The lowest levels were observed in the fractions of dichloromethane. In addition, **B4** contained the highest flavonoid content (715.553 ± 25.349 µg EQ/mg of extract), and the lowest flavonoid content was found in **B3** (10.279 ± 3.515 µg EQ/mg of extract). The results of the condensed tannin dosage show that only methanol extracts had condensed tannins, with amounts estimated at approximately two (02) times more in **B4** than in **B2** (223.770 ± 6.375 and 143.053 ± 2.674 µg EC/mg of extract, respectively). The negative values obtained with the dichloromethane extracts show that the latter do not contain condensed tannins. These results agree with those of the phytochemical screening, indicating the absence of this family of compounds in **B1** and **B3**. Therefore, we can conclude that dichloromethane is not an organic solvent suitable for the extraction of polar compounds such as tannins.

### 2.3. Antioxidant Content (AOC) of Extracts

Antioxidant activity was assessed (Figure 2) using the FRAP and DPPH methods. Trolox calibration curves were established by measuring the absorbance at 517 and 595 nm (for the DPPH and FRAP methods, respectively) based on the concentration of Trolox.

Figure 2 documents that methanolic extracts (**B2** and **B4**) of *A. macrostachya* have remarkable antioxidant content regardless of the method used. Fraction **B4** contains the highest antioxidant content, with Trolox equivalents estimated at 646.06 and 428.08 µg ET/mg of raw extract by the DPPH and FRAP methods, respectively. However, dichloromethane extracts **B1** and **B3** possessed no relevant antioxidant properties.

### 2.4. Cytotoxic Screening

Phytochemical analysis of the extracts revealed the presence of certain phytochemical families, such as flavonoids and alkaloids. Moreover, the literature has reported that the presence of these chemical groups in plant extracts confers pharmacological properties, such as anti-cancer properties. Therefore, to confirm this hypothesis and identify the extracts with major biological effects, the cytotoxicity of the extracts was assessed at 50 µg/mL (Figure 3, Figure 4 and Figure 5).

**B1** and **B2** strongly inhibited the growth of the two cell lines U937 and K562, indicating a combined cytostatic and cytotoxic effect of the extracts (Figure 3 and Figure 4) as potent as the antileukemic compound etoposide (VP16; Appendix A). **B4** had significant cytostatic/cytotoxic activity on U937 cells and moderate activity on K562 cells compared to the negative control (DMSO). **B3** had virtually no significant effects on either cell line compared to **B1**, **B2**, and **B4**. The root bark extracts showed higher cytotoxicity than the stem bark extracts. Additionally, the antiproliferative effects of the extracts were time-dependent (Figure 4).

Apoptosis induction by the different extracts was estimated by quantifying the percentage of cells showing apoptotic nuclear fragmentation (Appendix A) and confirmed by mitochondrial membrane potential loss analysis with two different methods (Hoechst and MitoTracker Red staining) (Figure 5).

Regardless of the method used, **B1** and **B2** had remarkable apoptotic effects on both types of cancer cells (U937 and K562) compared to the positive control, the chemotherapeutic agent etoposide (VP16; Appendix A). **B4** was selectively active on U937 cell lines with highly significant cytotoxicity. Its toxicity on K562 was moderate. **B3** did not contain proven cytotoxic substances. Its apoptotic activity was similar to that of the negative control (DMSO).

Cytotoxic screening, therefore, revealed that all extracts except **B3** had proven antiproliferative and apoptotic effects. Considering the results, subsequent analyses are directed to extracts with significant biological effects, specifically **B1**, **B2**, and **B4**.

### 2.5. Dose-Dependent Antiproliferative Effects of Extracts

Antiproliferative activity of the selected extracts (**B1**, **B2**, and **B4)** of *A. macrostachya* was assessed on chronic (K562) and acute (U937) myeloid leukemia cells and in parallel on proliferating PBMCs (pPBMCs) derived from healthy donors. The IC_50_ values are summarized in Table 1.

The analysis of the results in Table 1 reveals that **B1** and **B2** significantly inhibited the proliferation of U937 and K562 cells. **B1** showed the most remarkable antiproliferative activity, with IC_50_ values ranging from 2.7 ± 0.1 to 3.5 ± 1.5 µg/mL and from 6.0 ± 0.9 to 6.7 ± 0.7 µg/mL in U937 and K562 cells, respectively, with a maximum inhibition rate that varied depending on the concentration of the extract and the incubation time (24, 48, and 72 h). **B4** had no significant inhibitory effect on K562. The antiproliferative effects of different extracts were dose- and time-dependent (Figure 6). Moreover, unlike cancer cells, the extracts moderately inhibited proliferating PBMCs from healthy donors (Figure 7), with IC_50_ values ranging from 33.1 ± 10.6 to 40.1 ± 6.1 µg/mL for **B1** and greater than 50 µg/mL for **B2** and **B4**.

### 2.6. Dose-Dependent Apoptotic Effects of Different Extracts

The apoptotic effects of **B1**, **B2,** and **B4** were investigated in two cancer cell lines (K562 and U937) vs. proliferating PBMCs (pPBMCs). Apoptotic cell death was quantified by estimating the percentage of cells displaying apoptotic condensed/fragmented nuclei and confirmed by monitoring the percentage of cells showing mitochondrial membrane potential loss (by Hoechst and MitoTracker Red staining methods, respectively). The IC_50_ values of raw extracts of *A. macrostachya* against cancer cells (U937 and K562) as well as healthy proliferating cells (pPBMCs) are presented in Table 2.

Table 2 shows that extracts **B1** and **B2** were cytotoxic to both cancer cell models K562 and U937, with **B1** being more effective than **B2** overall. In contrast, **B4** selectively impacted U937 cells but not K562 cells (Figure 8 and Figure 9). **B1** possessed the highest pro-apoptotic effect, with IC_50_ values that ranged from 5.4 ± 0.2 to 6.0 ± 0.1 µg/mL in U937 cells and from 8.1 ± 2.1 to 8.7 ± 1.1 µg/mL in K562 cells. Overall, the extracts exhibited a dose-dependent pro-apoptotic effect (Figure 8 and Figure 9). **B1** showed the highest toxicity in healthy cell models (Figure 10). Nevertheless, the cytotoxicity of the **B1** extract on pPBMCs remained approximately 7 to 10 times less than that on cancer cells (Table 2), while **B2** and **B4** showed very moderate cytotoxicity against pPBMCs, as confirmed by the IC_50_ values (>50 µg/mL).

In short, all the results obtained indicate that **B1** and **B2** had highly significant antiproliferative and pro-apoptotic effects on both cancer cell lines. In addition, **B4** had cytotoxic activity on U937 cells and very moderate activity on K562 cells. Finally, **B3** had no remarkable cytotoxic effects on any of the cell lines.

## 3. Discussion

This study focused on four different extracts from root and stem barks of *Acacia macrostachya*, a medicinal plant used in Burkina Faso to treat many pathologies, including inflammation and cancer.

Phytochemical screening of the extracts revealed flavonoids, tannins, terpenoids/steroids, and alkaloids. All these observations are consistent with the bibliographic data collected. Therefore, the presence of these secondary metabolites could justify the cytotoxicity of this plant and its traditional use against specific pathologies. Indeed, it has already been demonstrated that the antioxidant and anti-cancer activities of plant extracts are linked to these phytochemical compounds [18,19,20,21,22,23,24,25].

A review of the TPC results revealed that **B4** contains more phenolic compounds. This result is consistent with those of previous chemical screenings that showed this family of compounds in the extract. The low level of phenolic compounds in the dichloromethane extracts (**B1** and **B3**) could be explained by the absence of chemical groups, including tannins, in these extracts due to their insolubility in apolar solvents. The content of phenolic compounds and total flavonoids of an extract depends strongly on the nature of the extraction solvent used. The higher the polarity of the solvent, the better the content of these compounds in the extracts with this solvent.

The methanol extract of stem bark (**B4**) showed the most interesting antioxidant activity. These results can be explained by the presence of phenolic compounds, especially flavonoids, in the different fractions that act through their OH^−^ groups capable of yielding electrons. All the results obtained are consistent with the literature [26,27,28,29] and demonstrate a good correlation between the contents of total phenolic compounds and flavonoids and their antioxidant activities. Thus, the differences in antioxidant capacities of *A. macrostachya* extracts could be explained by differences in the concentrations of polyphenols, particularly flavonoids.

Dichloromethane extract (**B1**) from root bark showed antiproliferative activity and a highly significant pro-apoptotic effect. This activity could be justified by the synergy of action of secondary compounds, such as terpenoids/steroids, flavonoids, and alkaloids, contained in the extract. Indeed, alkaloids have well-known anti-cancer properties. For example, some alkaloids, such as vinblastine and vincristine, isolated from extracts of tropical Madagascar periwinkle *Catharanthus roseus,* are anticancer agents currently used in the treatment of lung and breast cancer, as well as lymphoma [30,31,32,33]. In addition, vincristine is used against acute lymphoblastic leukemia, Hodgkin’s lymphoma, and non-Hodgkin’s lymphomas. Sawadogo et al. showed in 2011 that an *A. macrostachya* methanol extract significantly inhibited the proliferation of KB cancer cells and Vero and MCR-5 normal cells, with inhibition rates at 10 µg/mL of 95, 70, and 75%, respectively [2]. They correlated the cytotoxic activity of the extract with the presence of phytochemicals such as flavonoids. The same authors showed in 2015 that 5-hydroxyl-6,7,3′,4′,5′-pentamethoxyflavone isolated from *L. ukambensis* had relevant antiproliferative and pro-apoptotic effects against U937 [34]. In addition, curcumin, a polyphenolic molecule extracted from the roots of *Curcuma longa*, is capable of inducing death of K562 and Jurkat leukemia cells by apoptosis [35,36,37]. Catechins and procyanidins, molecules of the flavonoid family, have inhibitory effects on different carcinogenic processes [38]. Other authors revealed that rare derivatives of diterpenes isolated from sponges (*Spongionella sp.*) were found to have cytotoxic activity [39]. These substances tested on mononucleated peripheral blood cells were less cytotoxic than K562 leukemia cells. Furthermore, heteronemin, a sesterterpene isolated from *Hyrtios sp.*, can inhibit the activation of NF-κB and induce apoptosis in K562 cells [40].

Overall, we observed that the root bark extracts (**B1** and **B2**) showed better results than those extracted from the stem. The different chemical compositions could be one reason. The chemical composition may significantly vary from one region to another of the same plant. **B1** and **B2** would contain more compounds with interesting biological properties because of their tissue origin. Thus, dual cytostatic and cytotoxic effects appear. Overall, the number of both trypan blue-negative and trypan blue-positive cells indicated that the extracts exhibiting a stronger impact were able to completely block cell proliferation. Globally, we found the same number of cells seeded at time 0 h, i.e., after 24 h of treatment. Altogether, these observations suggest the strong cytotoxic effect of **B1** and **B2** as critical determinants to explain the potent cytostatic impact. Further studies will be required to elucidate the molecular mechanisms specifically involved and validate both in vitro and in vivo the anti-cancer potential of these extracts (or their single isolated compounds) to fulfill the long and multistep drug development process.

These results show the interest of *A. macrostachya* in the isolation of new anticancer molecules and constitute a scientific basis for the traditional uses of this plant.

## 4. Materials and Methods

### 4.1. Plant Material and Extraction

The root and stem barks of *Acacia macrostachya* were harvested in February 2018 at Laongo, a locality located north of Ouagadougou, capital of Burkina Faso, GPS coordinates 12°31′50, 52″ N; 01°17′2, 7″ W, and dried at room temperature for two weeks under ventilation. A voucher specimen was deposited in the herbarium of the University Joseph KI-ZERBO (identification number 17252). Dry plant material was transformed into a fine powder using an electric grinder (IRSAT, Ouagadougou, Burkina Faso). For each extraction, 50 g of plant powder was macerated in 150 mL of dichloromethane or methanol for 24 h under magnetic agitation. After filtration, the solvent was evaporated using a rotary (BUCHI Labortechnik GmbH, Hendrik-Ido-Ambacht, The Netherlands).

In total, four different extracts were obtained: dichloromethane extract from root bark (**B1**), methanol extract from root bark (**B2**), dichloromethane extract from stem bark (**B3**), and methanol extract from stem bark (**B4**). They were kept cool for the rest of the work.

For phytochemical analysis, such as spectrophotometric reading, all extracts were dissolved in methanol. Thus, each absorbance measurement corresponded to the average of three independent experiments.

### 4.2. Phytochemical Screening

Phytochemical screening consisted of qualitatively identifying the main chemical groups contained in each extract. In this work, flavonoids, tannins, alkaloids, and terpenoids/steroids were assessed. The following tube characterization methods were performed:Iron chloride test (III) for tannins;Aluminum chloride test for flavonoids;Acetic anhydride and sulfuric acid test for terpenoids/steroids;Potassium iodobismuthate test for alkaloids (Dragendorff method).

In addition, to confirm the presence of the chemical groups sought, thin layer chromatography (TLC) was carried out using silica gel F_254_ plates as a stationary phase with a fluorescent indicator at 254 nm, supporting appropriate elution systems. Then, chloroform-ethyl acetate (60/40 *v*/*v*) and hexane-ethyl acetate (20/4 *v*/*v*) were used as mobile phases for the flavonoids and the terpenoids/steroids, respectively. **B1** and **B3** were dissolved in dichloromethane, while **B2** and **B4** were dissolved in methanol. At the end of the migration, plates were exposed to UV light at 254 and 365 nm and, in some cases, sprayed with the Neu or Liebermann–Burchard reagent. Under ultraviolet (UV) light at 365 nm, flavonoid spots appeared yellow, green, orange, or fluorescent blue [41]. The Liebermann–Burchard reagent characterizes the presence of terpenoids and/or steroids in extracts through different colors, such as green and purple [42].

### 4.3. Dosage of Total Phenolic Content

The content of phenolic compounds was determined using the Folin–Ciocalteu reagent (FCR). In the alkaline medium, polyphenols reduce the heteropoly–phosphotungsates–molybdates in the FCR to a blue chromogen measured at 765 nm [43] proportional to the amount of phenolic compounds present in the sample. To 60 µL of each suitably diluted extract, 60 µL of FCR was added. After 6 min of incubation, 120 µL of 7.5% Na_2_CO_3_ (*m*/*v*) was added. The reaction mixture was then incubated at laboratory temperature for 1 h. The absorbance of all samples was measured at 765 nm against a blank using a spectrophotometer (SPECTROstar NANO, BMG LABTECH, Ortenberg, Germany). A standard calibration curve was drawn using gallic acid (y = 88.309x + 0.047; R^2^ = 0.999). The average of three readings was used, and the results were expressed in µg equivalent of gallic acid per mg of raw extract (µg EGA/mg of extract).

### 4.4. Dosage of Flavonoids

The total flavonoids of the different extracts were dosed according to Zhishen et al. with some modifications [44]. A total of 50 µL of each suitably diluted sample was mixed with 150 µL of distilled water, followed by 15 µL of NaNO_2_ to 5% (*m*/*v*). Five minutes later, 15 µL of AlCl_3_ 10% (*m*/*v*) was added. The reaction mixture was then incubated at room temperature for 6 min. Then, 50 µL of NaOH (1 N) was added, and the absorbances were immediately read at 510 nm against a blank using the spectrophotometer mentioned above. A calibration curve was established using quercetin as a reference in the same modus operandi as the samples. The flavonoid content of the sample, expressed in µg equivalent of quercetin per mg of raw extract (µg EQ/mg of extract), was obtained by reporting the absorbance read on the quercetin calibration curve used as a standard (y = 4.867x + 0.016; R^2^ = 0.999).

### 4.5. Dosage of Tannins

The dosage of condensed tannins was determined for the different extracts using the Broadhurst and Jones method with some modifications [45]. To 400 µL of each sample, 3 mL of the vanillin solution (4% in methanol) and 1.5 mL of concentrated hydrochloric acid (HCl) were added. After 15 min of reaction, the absorbances were read at 500 nm using the spectrophotometer mentioned above. The levels of condensed tannins were determined using the equation of the catechin calibration curve used as the standard (y = 23.058x + 0.031; R^2^ = 0.999). The results were expressed in µg equivalent of catechin per mg of raw extract (µg EC/mg of extract).

### 4.6. Evaluation of Antioxidant Activity

The antioxidant content of the different extracts of *Acacia macrostachya* was estimated using two methods: DPPH and FRAP. The DPPH method measures the ability of an extract to reduce the chemical radical DPPH^●^ (2.2-diphenyl-1-picrylhydrazyl) by transferring a proton to it. The discoloration of the initially purple solution was proportional to the amount of antioxidants in the sample. The FRAP method measured the ability of a compound to reduce the iron complex (Fe III) 2.4.6-tripyridyl-s-triazine (TPTZ) into a ferrous complex (Fe II) responsible for intense blue coloration, which was proportional to the antioxidant power of the dosed sample. Briefly, 50 µL of the diluted solution of the extract was mixed with 200 µL of the DPPH^●^ or FRAP reagent. After 10 min of incubation at laboratory temperature in the dark, the absorbances were read at 517 and 595 nm, respectively, by the DPPH method and the FRAP method using a spectrophotometer (SPECTROstar NANO, BMG LABTECH, Ortenberg, Germany). These optical densities were reported on the established calibration curves using Trolox (Sigma–Aldrich, St. Louis, MI, USA) as a reference antioxidant and prepared under the same conditions as extracts. The antioxidant content was estimated using the equations of the calibration curves y = −53.485x + 0.482; R^2^ = 0.997 and y = 135.31x + 0.282; R^2^ = 0.999 for the DPPH and FRAP methods, respectively. The results were expressed in µg equivalent of Trolox per mg of raw extract (µg ET/mg of extract). All measurements were repeated three times.

### 4.7. Cell Models

Acute myeloid leukemia U937 and chronic myelogenous leukemia K562 were purchased from the Deutsche Sammlung von Mikroorganismen und Zellkulturen (DSMZ, Braunschweig, Germany). Cells were cultured in RPMI 1640 medium (Lonza, Verviers, Belgium) supplemented with 10% (*v*/*v*) fetal calf serum (FCS; Sigma–Aldrich, Borneum, Belgium) and 1% (*v*/*v*) antibiotic/antimycotic (penicillin, streptomycin, and amphotericin; BioWhittaker, Verviers, Belgium) at 37 °C and 5% CO_2_. Cells were routinely checked for mycoplasma contamination. Experiments were performed on cells in exponential growth in culture medium containing 10% (*v*/*v*) FCS. Cells were seeded at a concentration of 200,000 cells/mL one hour before treatments. Peripheral blood mononuclear cells (PBMCs) were isolated from freshly collected buffy coats of healthy donors by density gradient centrifugation using Ficoll–Hypaque (GE Healthcare, Roosendaal, The Netherlands) as previously described [46]. Three buffy coats were kindly donated by Red Cross Luxembourg, with the approval of the Red Cross Luxembourg ethical committee and after obtaining written informed consent from donors. After purification, noncycling PBMCs were cultured overnight in RPMI 1640 supplemented with 10% (*v*/*v*) FCS and 1% (*v*/*v*) antibiotic/antimycotic at 37 °C and 5% CO_2_ at a cell density of 2.0 × 10^6^/mL. One day later, their concentration was adjusted to 1.0 × 10^6^/mL using prewarmed complete culture medium, and then treatments were added after one hour. In parallel, 1.0 × 10^8^/mL PBMCs were induced to proliferate as previously described [46]. Briefly, after Ficoll–Hypaque purification, PBMCs were incubated for at least one hour at 37 °C, and 5% CO_2_ in a flask preconditioned with 10 mL of inactivated human AB serum (Corning, Manassas, VA, USA) at a cell density of 1.0 × 10^6^/mL in RPMI 1640 supplemented with 10% (*v*/*v*) inactivated human AB serum and 1% (*v*/*v*) antibiotic/antimycotic. At the end of the incubation, the floating cells corresponding to the enriched lymphoid population were harvested, pelleted by centrifugation, and resuspended at a cell density of 1.0 × 10^6^/mL in RPMI 1640 supplemented with 1% (*v*/*v*) antibiotic/antimycotic, 1% (*v*/*v*) HEPES (Invitrogen Tournai, Belgium), 1% (*v*/*v*) sodium pyruvate (Invitrogen), 1% nonessential amino acids (Invitrogen), 0.1% 2-mercaptoethanol (Invitrogen), 5% (*v*/*v*) FCS, 10% (*v*/*v*) human AB serum, 1 µg/mL phytohemagglutinin (PHA, Gentaur, Kampenhout, Belgium) and 50 µg/mL interleukin-2 (IL-2; Roche, Luxembourg, Luxembourg). After 72 h of culture at 37 °C and 5% CO_2_, the blastogenic response and T cell enrichment were validated as previously described [46]; then, the cell concentration was adjusted to 1.0 × 10^6^/mL before treatments.

### 4.8. Evaluation of Cytotoxicity

#### 4.8.1. Trypan Blue Viability Assay

The antiproliferative and cytotoxic effects of the extracts were assessed by counting the number of viable cells per mL by the trypan blue exclusion test [46]. The cell concentration and the percentage of trypan blue-positive cells were estimated using an automated system (Cedex XS, Roche Diagnostics, Mannheim, Germany). U937 and K562 cells were seeded in a 6-well plate (cell density: 200,000 cells/mL; 2 mL/well) [34]; proliferating PBMCs were seeded in a 12-well plate (cell density: 1.0 × 10^6^/mL; 1 mL/well). After one hour, cells were challenged with the indicated range of concentrations of the raw extracts. Dried extract powders were dissolved in dimethyl sulfoxide (DMSO) at the concentration of 5 mg/ml. This stock was used to generate all the working concentrations. The same volume of vehicle dimethyl sulfoxide (DMSO) was added to the untreated cells as a negative control. As a positive control, cells were incubated with the chemotherapeutic agent etoposide (VP16, 30 µM; Sigma–Aldrich) for the indicated times. In all experiments of this study, the volume of DMSO used did not exceed 0.25% to avoid any significant solvent toxicity on the tested cells. At regular time intervals (24, 48, and 72 h), 20 µL of the cell suspension was added to 20 µL of trypan blue solution (Roche Diagnostics) [34,46]. Twenty microliters of the mixture were deposited on a compatible Cedex XS slide. Values corresponding to the viable cell concentration (number of cells × 10^5^/mL) and the percentage of trypan blue-positive cells were recorded with computer-controlled software (Cedex XS Software, Roche Diagnostic). Antiproliferative and cytotoxic IC_50_ values were estimated using GraphPad Prism software (8.3.0 version (GraphPad, La Jolla, CA, USA)).

#### 4.8.2. Hoechst and MitoTracker Red Staining Methods

The analysis of apoptotic effects of the plant extracts was performed using two methods described by Sawadogo et al., with some modifications [34,47]: analysis of nuclear fragmentation/condensation by fluorescence microscopy (Hoechst staining method) and analysis of the mitochondrial membrane potential by flow cytometry (MitoTracker Red staining method). Hoechst 33342 (Molecular Probes, Invitrogen) is a cell-permeant dye that binds to chromatin and allows the monitoring of nuclear morphology. At the indicated times, 1 mL of cells was incubated for 15 min at 37 °C and 5% CO_2_ with 1 µg/mL Hoechst 33342. At the end of the incubation, the percentage of cells with apoptotic fragmented/condensed nuclei was estimated by counting cells with nuclear apoptotic morphology over the total in at least three random fields of 300 cells (Equation (1)) using a fluorescence microscope (Olympus, Hamburg, Germany). Cell-permeant MitoTracker Red (MTR; Invitrogen, Molecular Probes) accumulates in the mitochondria depending on the mitochondrial membrane potential. Living cells accumulate MTR dye in mitochondria, resulting in a peak in intensity, while apoptotic cells exhibit reduced fluorescence intensity. After exposure to the indicated concentrations of the extracts for 24 and 48 h, 500 µL of cells were incubated for 15 min at 37 °C and 5% CO_2_ with 100 nM MTR. At the end of the incubation, the percentage of cells with mitochondrial membrane potential loss was estimated (Equation (2)) using a FACSCalibur (Becton Dickinson, San José, CA, USA) tuned at 488 nm (emission: 585 nm; FL3-H: 625 nm). A total of 10,000 events were recorded using Cell Quest software (Becton Dickinson, San José, California), and further analysis was performed using FlowJo 10 software (Tree Star Inc., Ashland, OR, USA).
(1)$$\%\;\mathrm{cell}\;\mathrm{death}=\frac{\mathrm{Number}\;\mathrm{of}\;\mathrm{cells}\;\mathrm{with}\;\mathrm{fragmented}/\mathrm{condensed}\;\mathrm{nuclei}}{\mathrm{Total}\;\mathrm{number}\;\mathrm{of}\;\mathrm{cells}}\times 100$$
(2)$$\%\;\mathrm{cell}\;\mathrm{death}=\frac{\mathrm{Number}\;\mathrm{of}\;\mathrm{cells}\;\mathrm{with}\;\mathrm{mitochondrial}\;\mathrm{membrane}\;\mathrm{potential}\;\mathrm{loss}}{\mathrm{Total}\;\mathrm{number}\;\mathrm{of}\;\mathrm{cells}}\times 100$$

### 4.9. Statistical Analysis

The results are expressed as the mean ± SD of at least three independent experiments. Statistical analyses were performed using one-way or two-way ANOVA as indicated, followed by appropriate post hoc tests. A value of *p* < 0.05 was considered statistically significant. Graphs were generated with GraphPad Prism software version 8.3.0 (GraphPad, La Jolla, CA, USA).

## 5. Conclusions

In this study, we assessed the differential cytotoxicity induced by two different extracts from the root and the stem bark of *Acacia macrostachya* in U937 and K562 leukemia cells compared to proliferating non-cancerous PBMCs from healthy individuals. Our results document a selective antiproliferative and proapoptotic activity of **B1**, **B2**, and **B4** extracts in both cancer cell lines; in contrast, noncancer proliferating cell models were not or only very moderately impacted. Based on previously published results, this is the first time that different extracts from the roots and stem barks of the *Acacia macrostachya* plant have been investigated for their cytotoxic activity: (1) integrating the information derived from the individual variability of ex vivo healthy specimens (proliferating PBMCs); and (2) using a multiparametric analysis (plasma membrane integrity, mitochondrial membrane potential loss, and nuclear morphology analysis). However, the use of extracts and the limited number of cell models tested represent the limitations of the present study. In our future investigations, a bioguided split will be performed to isolate, identify, and characterize the anticancer molecules of *A. macrostachya*. Furthermore, based on preliminary in vitro/ex vivo results, we plan to extend our analysis to a broader panel of cancer cell models and perform in vivo studies.

## Figures and Tables

**Figure 1 plants-10-01353-f001:**
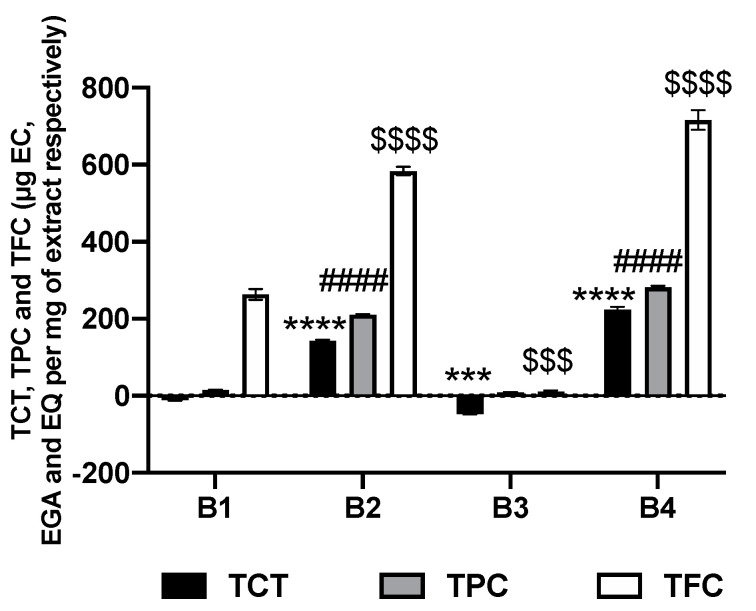
Total phenolic content (TPC), total flavonoid content (TFC), and total condensed tannins (TCT) in *A. macrostachya* extracts; values are given as the mean ± SD of three (03) independent assays. **B1**: dichloromethane extract from root bark; **B2**: methanol extract from root bark; **B3**: dichloromethane extract from stem bark; **B4**: methanol extract from stem bark. Data were analyzed using two-way ANOVA; post hoc: Tukey: ***; $$$: *p* < 0.001; ****; ####; $$$$: *p* < 0.0001 (comparative analysis between **B1** and the other extracts).

**Figure 2 plants-10-01353-f002:**
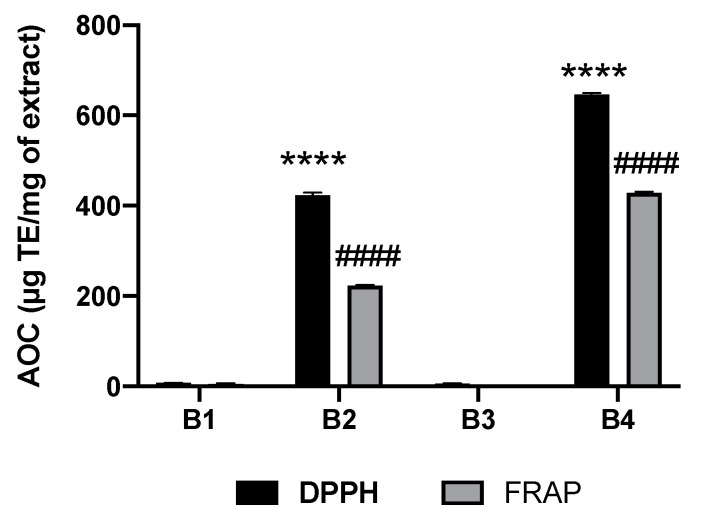
Antioxidant content of *A. macrostachya* extracts; three (03) independent assays were performed, and the standard deviation was calculated. **B1**: dichloromethane extract from root bark; **B2**: methanol extract from root bark; **B3**: dichloromethane extract from stem bark; **B4**: methanol extract from stem bark. Data were analyzed using two-way ANOVA; post hoc: Dunnett: ****; ####: *p* < 0.0001 (comparative analysis between the different extracts).

**Figure 3 plants-10-01353-f003:**
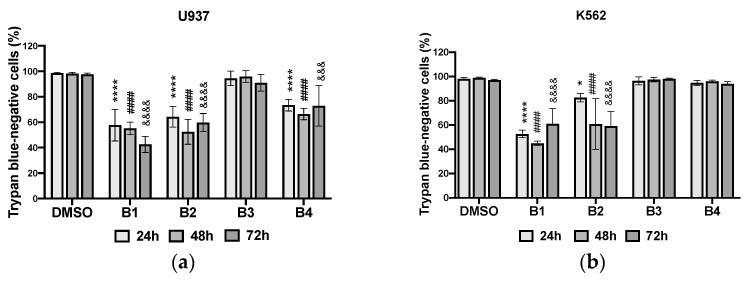
Cytotoxic effects of different extracts at 50 µg/mL and DMSO (used as negative control) on U937 (**a**) and K562 (**b**) cells after 24, 48, and 72 h. **B1**: dichloromethane extract from root bark; **B2**: methanol extract from root bark; **B3**: dichloromethane extract from stem bark; **B4**: methanol extract from stem bark. Data were analyzed using two-way ANOVA; post hoc: Dunnett: *: *p* < 0.05; &&&: *p* < 0.001; ****; ####; &&&&: *p* < 0.0001.

**Figure 4 plants-10-01353-f004:**
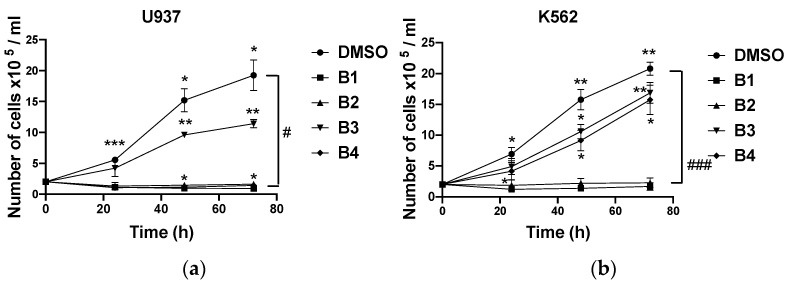
Antiproliferative effects of different extracts at 50 µg/mL and DMSO (used as negative control) on U937 (**a**) and K562 (**b**) cells after 24, 48, and 72 h. **B1**: dichloromethane extract from root bark; **B2**: methanol extract from root bark; **B3**: dichloromethane extract from stem bark; **B4**: methanol extract from stem bark. Data were analyzed using two-way ANOVA; post hoc: Dunnett/Tukey: *; #: *p* < 0.05; **: *p* < 0.01; ***; ###: *p* < 0.001.

**Figure 5 plants-10-01353-f005:**
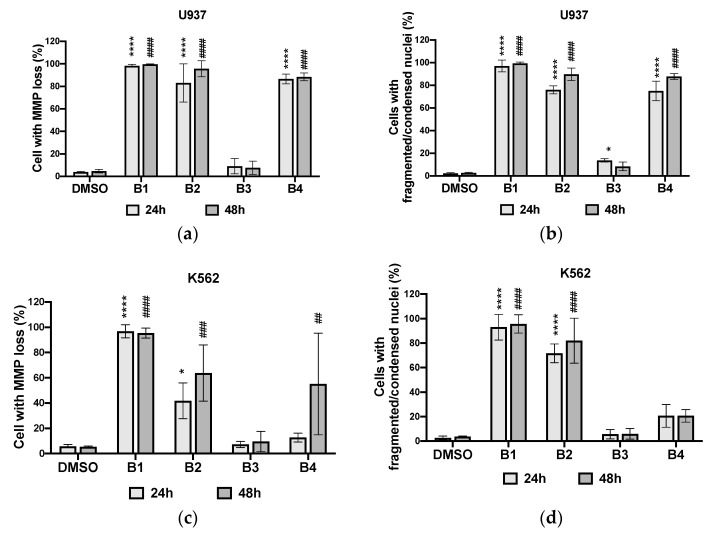
Apoptotic effects of different extracts at 50 µg/mL and DMSO (used as negative control) on U937 (**a**,**b**) and K562 (**c**,**d**) cells by Hoechst and MitoTracker Red staining analysis for 24 and 48 h. **B1**: dichloromethane extract from root bark; **B2**: methanol extract from root bark; **B3**: dichloromethane extract from stem bark; **B4**: methanol extract from stem bark. Data were analyzed using two-way ANOVA; post hoc: Dunnett: *: *p* < 0.05; ##: *p*< 0.01; ###: *p* < 0.001; ****; ####: *p* < 0.0001.

**Figure 6 plants-10-01353-f006:**
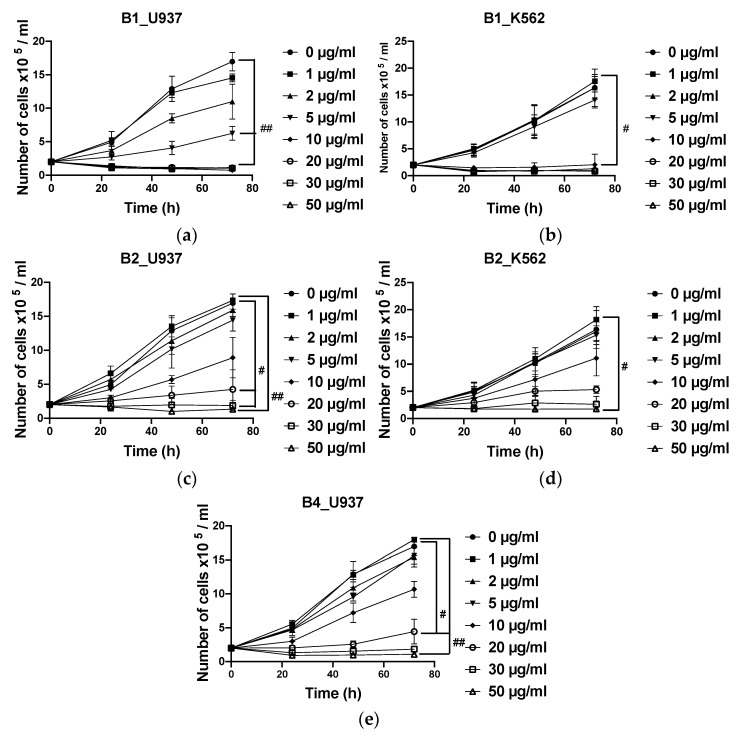
Antiproliferative effects of extracts at different concentrations and DMSO (used as negative control) on U937. (**a**,**c**,**e**) and K562 (**b**,**d**) cells after 24, 48, and 72 h. Data were analyzed using two-way ANOVA; post hoc: Dunnett: #: *p* < 0.05; ##: *p*< 0.01.

**Figure 7 plants-10-01353-f007:**
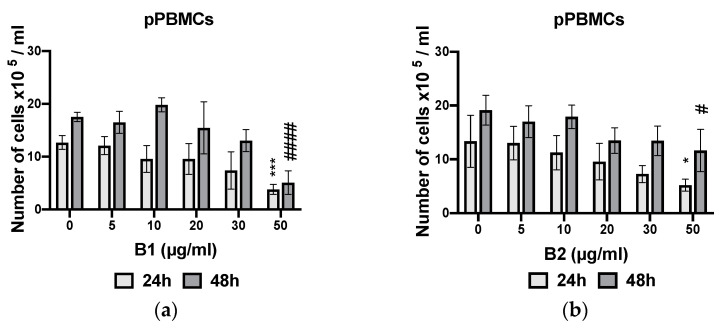
Antiproliferative effects of extracts at different concentrations on proliferating peripheral blood mononuclear cells (pPBMCs) (**a**–**c**) after 24 and 48 h. Data were analyzed using two-way ANOVA; post hoc: Dunnett: *, #: *p* < 0.05; ***: *p* < 0.001, ####: *p* < 0.0001.

**Figure 8 plants-10-01353-f008:**
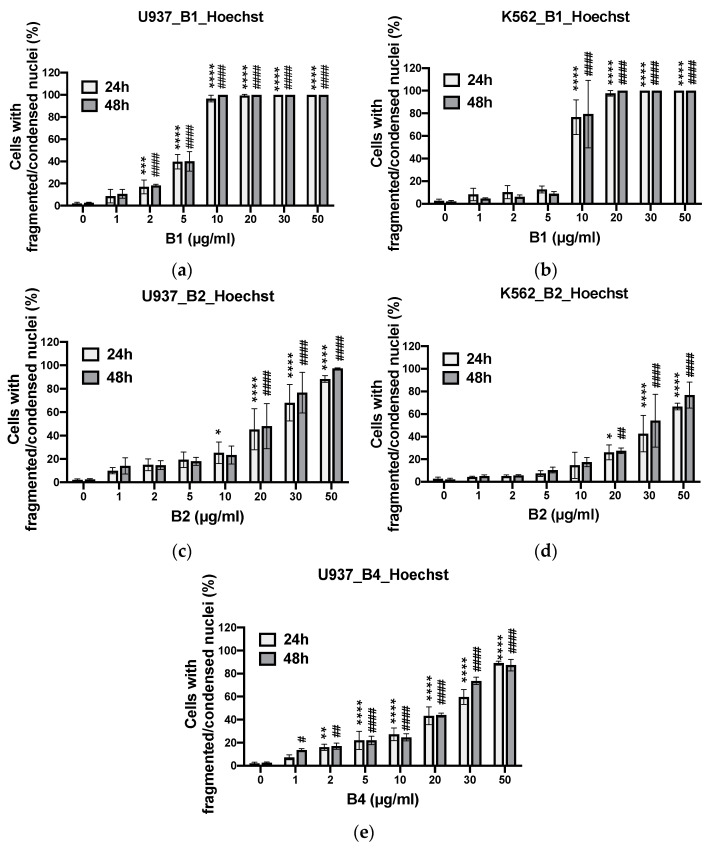
Apoptotic effects of extracts at different concentrations and DMSO (used as a negative control) on U937 (**a**,**c**,**e**) and K562 (**b**,**d**) cells using the Hoechst staining method for 24 and 48 h. Data were analyzed using two-way ANOVA; post hoc: Dunnett: *; #: *p* < 0.05; **; ##: *p* < 0.01; ***: *p* < 0.001; ****; ####: *p* < 0.0001.

**Figure 9 plants-10-01353-f009:**
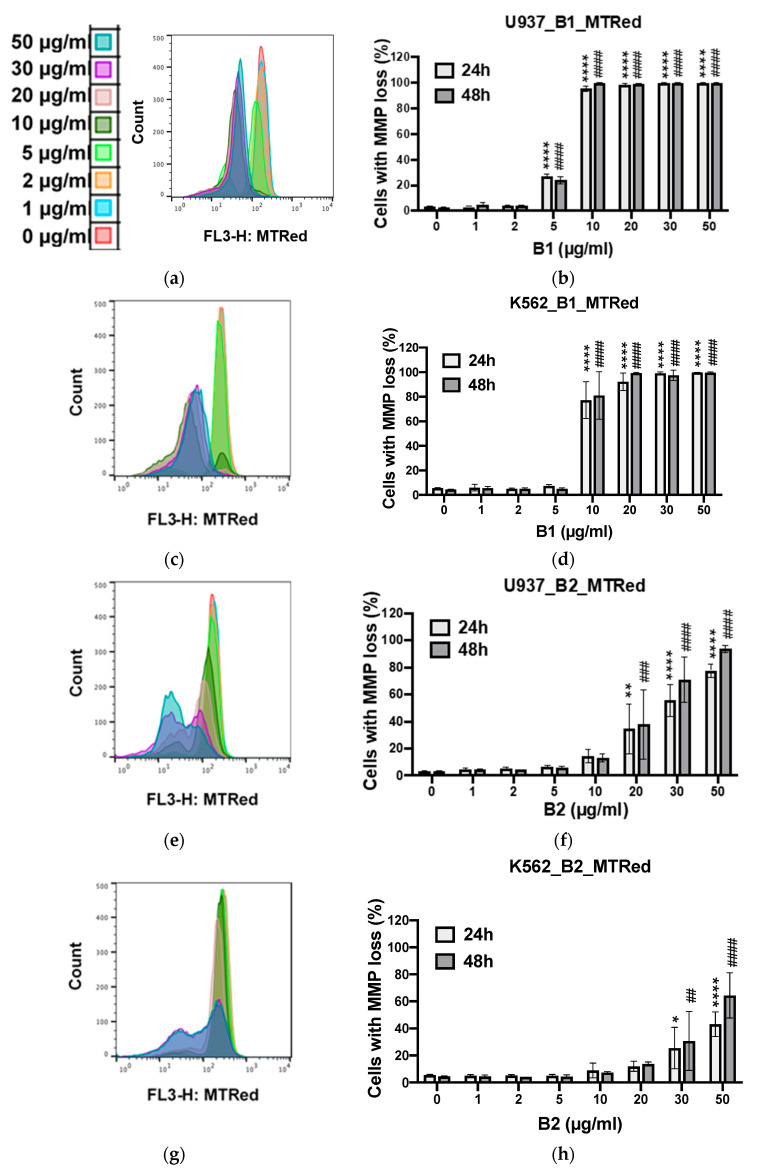
Apoptotic effects of extracts at different concentrations and DMSO (used as a negative control) on U937 (**b**,**f**,**j**) and K562 (**d**,**h**) cells using the MitoTracker Red staining method for 24 and 48 h; profile panels (**a**,**c**,**e**,**g**,**i**) provided by the flow cytometer. Data were analyzed using two-way ANOVA; post hoc: Dunnett: *: *p* < 0.05; **; ##: *p* < 0.01; ###: *p* < 0.001; ****; ####: *p* < 0.0001.

**Figure 10 plants-10-01353-f010:**
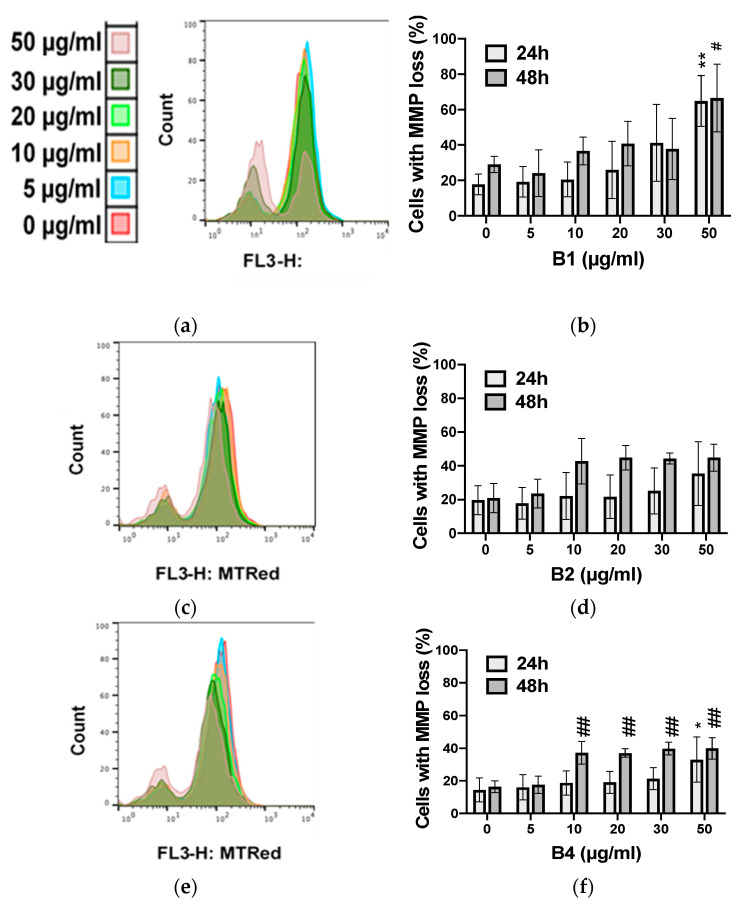
Dose- and kinetic-dependent apoptotic effects of extracts at different concentrations on proliferating peripheral blood mononuclear cells (pPBMCs) (**b**,**d**,**f**) using the MitoTracker Red staining method for 24 and 48 h; profile panels (**a**,**c**,**e**) provided by the flow cytometer. Data were analyzed using two-way ANOVA; post hoc: Dunnett: *; #: *p* < 0.05; **; ##: *p* < 0.01.

**Table 1 plants-10-01353-t001:** IC_50_ (µg/mL) values of the antiproliferative activity of different extracts of *A. macrostachya*.

IC_50_ (µg/mL)				
Cell Models	Hours	*A. macrostachya* Extracts		
		B1	B2	B4
	**24**	3.5 ± 1.5	6.9 ± 1.6	13.1 ± 6.0
**U937**	**48**	2.7 ± 0.1	9.8 ± 1.9	10.9 ± 1.7
	**72**	3.0 ± 0.7	13.6 ± 6.1	12.5 ± 1.2
	**24**	6.6 ± 1.3	13.7 ± 4.0	
**K562**	**48**	6.7 ± 0.7	13.6 ± 4.9	ND
	**72**	6.0 ± 0.9	12.5 ± 2.2	
**pPBMCs**	**24**	33.1 ± 10.6	>50	>50
	**48**	40.1 ± 6.1	>50	>50

ND: not determined.

**Table 2 plants-10-01353-t002:** IC_50_ (µg/mL) values of the apoptotic effect of different extracts of *A. macrostachya* on cancer cells compared to healthy cells.

IC_50_ (µg/mL)					
Methods	Cell Models	Hours	*A. macrostachya* Extracts		
			B1	B2	B4
**MitoTracker Red**	**U937**	**24**	6.0 ± 0.1	24.6 ± 6.2	20.2 ± 5.0
		**48**	5.5 ± 0.2	23.3 ± 6.5	22.0 ± 3.6
	**K562**	**24**	8.2 ± 1.0	32.4 ± 9.2	ND
		**48**	8.1 ± 2.1	30.6 ± 8.9	
**Hoechst**	**U937**	**24**	5.5 ± 0.2	25.9 ± 9.3	26.6 ± 1.3
		**48**	5.4 ± 0.2	26.7 ± 9.2	23.6 ± 1.3
	**K562**	**24**	8.4 ± 0.7	>50	ND
		**48**	8.7 ± 1.1	47.2 ± 4.9	
**MitoTracker Red**	**pPBMCs**	**24**	32.0 ± 5.7	>50	>50
		**48**	42.1 ± 16.7	>50	>50

ND: not determined.

## Data Availability

Data supporting reported results are available at: https://data.mendeley.com/datasets/dshfydf55n/1, accessed on 30 June 2021.

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
