# Peer review of "Phytochemical Screening and Antioxidant and Cytotoxic Effects of Acacia macrostachya"

_plants, 2021, doi:10.3390/plants10071353_

Round 1

Reviewer 1 Report

Dear authors,

I understand that you do not have available highly sophisticated methodology for research and that you are limited in performing metabolomic studies, but in this case, in my opinion, you should tray to publish your research in a journal with lower IF, not in Q1. As someone who has long year experiance in phytochemical study I can not recommend your work to be published in Q1 journal. 

Your study gives just results about cytotoxicity of two different types of extracts from root and stem bark of Acacia macrostachya against U937 and K562 cells. This are in vitro studies and may not have in vivo effects. You did not mention in your manuscript that from in vitro to in vivo is a long process. In my experiance, every tested extracts will show antioxidant and cytotoxic effect, to some extent, in in vitro studies, so I do not see in your work something worth of publishing in Q1 journal.

This plants is also very well studied about bioactivity and phytochemical content. Just quick literature search will gave a lots of published results.

Also I have concerns that you did not perform cell-test in a proper way. You say: "After one hour, cells were challenged with the indicated range of concentrations of the raw extracts". You prepaire raw extracts in different solvents, but you used as a negative control just DMSO. As a negative control solvent which you use for preparation of the extracts should be used, because also extract could influence final results.

In a most of the Tables and Figures you did not provide information about statistical significance.

Author Response

Reviewer #1

 General comments (reviewer #1)

Dear authors,

I understand that you do not have available highly sophisticated methodology for research and that you are limited in performing metabolomic studies, but in this case, in my opinion, you should tray to publish your research in a journal with lower IF, not in Q1. As someone who has long year experiance in phytochemical study I can not recommend your work to be published in Q1 journal. 

Your study gives just results about cytotoxicity of two different types of extracts from root and stem bark of Acacia macrostachya against U937 and K562 cells. This are in vitro studies and may not have in vivo effects. You did not mention in your manuscript that from in vitro to in vivo is a long process. In my experiance, every tested extracts will show antioxidant and cytotoxic effect, to some extent, in in vitro studies, so I do not see in your work something worth of publishing in Q1 journal.

We provide here an experimental basis exploring the pharmacological potential of a traditionally used plant. We oriented our investigations towards the cytostatic/cytotoxic effects. In addition to cancer cell models, our extracts did not adversely affect ex vivo proliferating PBMCs from healthy individuals PBMCs, in contrast to the known chemotherapeutic agent etoposide (VP16) used at therapeutic concentrations. Overall, our results confirm the potential of a traditionally used plant and encourage further progress. In the conclusion (section 5.), a paragraph mentioning the novel aspects and the limitations of the present study was inserted (lines 509-519). Furthermore, we mention the long and multi-step process of drug development at the end of the discussion (lines 324-326).

This plants is also very well studied about bioactivity and phytochemical content. Just quick literature search will gave a lots of published results.

A Pubmed search documents a total of 5 publications dealing with Acacia macrostachya. Only two studies include tests on cancer cells. One of these two studies correspond to the previous work of the authors at the Institut de Recherche en Sciences de la Santé, Ouagadougou-Burkina Faso. Both studies are limited to preliminary screening activities. Based on the publications available and our previous results, this is the first time that different extracts from the roots and barks of Acacia macrostachya plant have been investigated for their cytotoxic activity: 1) integrating the information derived from the individual variability of ex vivo healthy specimens (proliferating PBMCs); 2) using a multiparametric analysis (plasma membrane integrity, mitochondrial membrane potential loss and nuclear morphology analysis).

Also I have concerns that you did not perform cell-test in a proper way. You say: "After one hour, cells were challenged with the indicated range of concentrations of the raw extracts". You prepaire raw extracts in different solvents, but you used as a negative control just DMSO. As a negative control solvent which you use for preparation of the extracts should be used, because also extract could influence final results.

The reviewer can be assured that the “cell-test” were done properly according to the standards in use at LBMCC, Luxembourg and the College of Pharmacy of Seoul National University, South Korea. The author’s well-documented technological knowledge and cell biological approaches allowed to publish data in the field of molecular and cellular anticancer effects of natural extracts, compounds and hemi-synthetic derivatives in Q1 and top-10 journal in the field of oncology, hematology and pharmacology, amongst others [1-12]. Moreover, publication-relevant topics related to cell death and natural compounds were discussed in review papers with an IF >10 [13-20], further documenting the authors insight into the topic. Altogether the authors believe that the concerns of the reviewer are not justified.

Briefly, the extracts were dried and stored as a dried powder at +°4C in the dark. This procedure prevents the presence of methanol. For in cellulo analyses, the powder was dissolved in DMSO at a concentration of 5 mg/ml. This stock was used to generate working dilutions. A sentence has been added to Materials and Methods (section 4.8.1, lines 462-463). Accordingly. DMSO served as negative control solvent.

In a most of the Tables and Figures you did not provide information about statistical significance.

We carefully checked this point again. We could not identify any omission of statistical analysis in our figures. This applies to both main and supplementary Figures. In table 1 and 2, we report the average of IC50 values for the anti-proliferative/cytotoxic effects of the extracts in the different cell models.

Rebuttal references:

  1. Chateauvieux, S.; Gaigneaux, A.; Gerard, D.; Orsini, M.; Morceau, F.; Orlikova-Boyer, B.; Farge, T.; Recher, C.; Sarry, J.E.; Dicato, M., et al. Inflammation regulates long non-coding RNA-PTTG1-1:1 in myeloid leukemia. Haematologica 2020, 105, e280-e284, doi:10.3324/haematol.2019.217281.
  2. Orsini, M.; Chateauvieux, S.; Rhim, J.; Gaigneaux, A.; Cheillan, D.; Christov, C.; Dicato, M.; Morceau, F.; Diederich, M. Sphingolipid-mediated inflammatory signaling leading to autophagy inhibition converts erythropoiesis to myelopoiesis in human hematopoietic stem/progenitor cells. Cell Death Differ 2019, 26, 1796-1812, doi:10.1038/s41418-018-0245-x.
  3. Mazumder, A.; Lee, J.Y.; Talhi, O.; Cerella, C.; Chateauvieux, S.; Gaigneaux, A.; Hong, C.R.; Kang, H.J.; Lee, Y.; Kim, K.W., et al. Hydroxycoumarin OT-55 kills CML cells alone or in synergy with imatinib or Synribo: Involvement of ER stress and DAMP release. Cancer Lett 2018, 438, 197-218, doi:10.1016/j.canlet.2018.07.041.
  4. Lee, J.Y.; Talhi, O.; Jang, D.; Cerella, C.; Gaigneaux, A.; Kim, K.W.; Lee, J.W.; Dicato, M.; Bachari, K.; Han, B.W., et al. Cytostatic hydroxycoumarin OT52 induces ER/Golgi stress and STAT3 inhibition triggering non-canonical cell death and synergy with BH3 mimetics in lung cancer. Cancer Lett 2018, 416, 94-108, doi:10.1016/j.canlet.2017.12.007.
  5. Ji, S.; Lee, J.Y.; Schror, J.; Mazumder, A.; Jang, D.M.; Chateauvieux, S.; Schnekenburger, M.; Hong, C.R.; Christov, C.; Kang, H.J., et al. The dialkyl resorcinol stemphol disrupts calcium homeostasis to trigger programmed immunogenic necrosis in cancer. Cancer Lett 2018, 416, 109-123, doi:10.1016/j.canlet.2017.12.011.
  6. Yagdi Efe, E.; Mazumder, A.; Lee, J.Y.; Gaigneaux, A.; Radogna, F.; Nasim, M.J.; Christov, C.; Jacob, C.; Kim, K.W.; Dicato, M., et al. Tubulin-binding anticancer polysulfides induce cell death via mitotic arrest and autophagic interference in colorectal cancer. Cancer Lett 2017, 410, 139-157, doi:10.1016/j.canlet.2017.09.011.
  7. Schnekenburger, M.; Goffin, E.; Lee, J.Y.; Jang, J.Y.; Mazumder, A.; Ji, S.; Rogister, B.; Bouider, N.; Lefranc, F.; Miklos, W., et al. Discovery and Characterization of R/S-N-3-Cyanophenyl-N'-(6-tert-butoxycarbonylamino-3,4-dihydro-2,2-dimethyl-2H-1- benzopyran-4-yl)urea, a New Histone Deacetylase Class III Inhibitor Exerting Antiproliferative Activity against Cancer Cell Lines. J Med Chem 2017, 60, 4714-4733, doi:10.1021/acs.jmedchem.7b00533.
  8. Cerella, C.; Gaigneaux, A.; Mazumder, A.; Lee, J.Y.; Saland, E.; Radogna, F.; Farge, T.; Vergez, F.; Recher, C.; Sarry, J.E., et al. Bcl-2 protein family expression pattern determines synergistic pro-apoptotic effects of BH3 mimetics with hemisynthetic cardiac glycoside UNBS1450 in acute myeloid leukemia. Leukemia 2017, 31, 755-759, doi:10.1038/leu.2016.341.
  9. Radogna, F.; Cerella, C.; Gaigneaux, A.; Christov, C.; Dicato, M.; Diederich, M. Cell type-dependent ROS and mitophagy response leads to apoptosis or necroptosis in neuroblastoma. Oncogene 2016, 35, 3839-3853, doi:10.1038/onc.2015.455.
  10. Okoye, F.B.; Sawadogo, W.R.; Sendker, J.; Aly, A.H.; Quandt, B.; Wray, V.; Hensel, A.; Esimone, C.O.; Debbab, A.; Diederich, M., et al. Flavonoid glycosides from Olax mannii: Structure elucidation and effect on the nuclear factor kappa B pathway. J Ethnopharmacol 2015, 176, 27-34, doi:10.1016/j.jep.2015.10.019.
  11. Orhan, I.E.; Kartal, M.; Gulpinar, A.R.; Yetkin, G.; Orlikova, B.; Diederich, M.; Tasdemir, D. Inhibitory effect of St. Johns Wort oil macerates on TNFalpha-induced NF-kappaB activation and their fatty acid composition. J Ethnopharmacol 2014, 155, 1086-1092, doi:10.1016/j.jep.2014.06.030.
  12. El Amrani, M.; Lai, D.; Debbab, A.; Aly, A.H.; Siems, K.; Seidel, C.; Schnekenburger, M.; Gaigneaux, A.; Diederich, M.; Feger, D., et al. Protein kinase and HDAC inhibitors from the endophytic fungus Epicoccum nigrum. J Nat Prod 2014, 77, 49-56, doi:10.1021/np4005745.
  13. Cerella, C.; Teiten, M.H.; Radogna, F.; Dicato, M.; Diederich, M. From nature to bedside: pro-survival and cell death mechanisms as therapeutic targets in cancer treatment. Biotechnol Adv 2014, 32, 1111-1122, doi:10.1016/j.biotechadv.2014.03.006.
  14. Mazumder, A.; Cerella, C.; Diederich, M. Natural scaffolds in anticancer therapy and precision medicine. Biotechnol Adv 2018, 36, 1563-1585, doi:10.1016/j.biotechadv.2018.04.009.
  15. Morceau, F.; Chateauvieux, S.; Orsini, M.; Trecul, A.; Dicato, M.; Diederich, M. Natural compounds and pharmaceuticals reprogram leukemia cell differentiation pathways. Biotechnol Adv 2015, 33, 785-797, doi:10.1016/j.biotechadv.2015.03.013.
  16. Schnekenburger, M.; Dicato, M.; Diederich, M. Plant-derived epigenetic modulators for cancer treatment and prevention. Biotechnol Adv 2014, 32, 1123-1132, doi:10.1016/j.biotechadv.2014.03.009.
  17. Schumacher, M.; Kelkel, M.; Dicato, M.; Diederich, M. Gold from the sea: marine compounds as inhibitors of the hallmarks of cancer. Biotechnol Adv 2011, 29, 531-547, doi:10.1016/j.biotechadv.2011.02.002.
  18. Diederich, M.; Cerella, C. Non-canonical programmed cell death mechanisms triggered by natural compounds. Semin Cancer Biol 2016, 40-41, 4-34, doi:10.1016/j.semcancer.2016.06.001.
  19. Florean, C.; Dicato, M.; Diederich, M. Immune-modulating and anti-inflammatory marine compounds against cancer. Semin Cancer Biol 2020, 10.1016/j.semcancer.2020.02.008, doi:10.1016/j.semcancer.2020.02.008.
  20. Cerella, C.; Dicato, M.; Diederich, M. BH3 Mimetics in AML Therapy: Death and Beyond? Trends Pharmacol Sci 2020, 41, 793-814, doi:10.1016/j.tips.2020.09.004.

Reviewer 2 Report

The revised version has addressed most of the questions previously presented upon the first revision round. Since the dichloromethane extracts were solubilized in methanol for the phenolic content, flavonoids, DPPH, FRAP my previous concerns regarding the methodology used are rendered mute. The same applies to the units used, which as ug/mg make much more sense than previously.

There are only some minor errors or changes needed as stated below:

Line 24: FRAP stands for Ferric Reducing Antioxidant Power and not (fluorescence recovery after photobleaching). These methods are nothing alike…

Line 72: Change “and then limit water losses” for “by limiting water losses”

Figure B2 caption should also indicate what B1-B4 stand for as changed in all other Figures.

Author Response

Reviewer #2

General comments (reviewer #2)

The revised version has addressed most of the questions previously presented upon the first revision round. Since the dichloromethane extracts were solubilized in methanol for the phenolic content, flavonoids, DPPH, FRAP my previous concerns regarding the methodology used are rendered mute. The same applies to the units used, which as ug/mg make much more sense than previously.

We thank reviewer #2 for the valuable comments.

Specific comments (reviewer #2)

 There are only some minor errors or changes needed as stated below:

 Line 24: FRAP stands for Ferric Reducing Antioxidant Power and not (fluorescence recovery after photobleaching). These methods are nothing alike…

We apologize for the oversight. In the revised version, the definition of FRAP was corrected.

Line 72: Change “and then limit water losses” for “by limiting water losses”

In the revised version, the term “and then limit water losses” was replaced by “by limiting water losses”.

 Figure B2 caption should also indicate what B1-B4 stand for as changed in all other Figures.

We thank the reviewer 2 for this suggestion. This recommendation was implemented throughout the text.

Reviewer 3 Report

The manuscript should be published as presented.

Author Response

Reviewer #3

The manuscript should be published as presented.

We thank the reviewer 3 for the positive evaluation of our manuscript.

Round 2

Reviewer 1 Report

No additional comment.